# A Possible Way to Relate the Effects of SARS-CoV-2-Induced Changes in Transferrin to Severe COVID-19-Associated Diseases

**DOI:** 10.3390/ijms23116189

**Published:** 2022-05-31

**Authors:** Elek Telek, Zoltán Ujfalusi, Gábor Kemenesi, Brigitta Zana, Ferenc Jakab, Gabriella Hild, András Lukács, Gábor Hild

**Affiliations:** 1Department of Biophysics, Medical School, University of Pécs, Szigeti Str. 12, H-7624 Pécs, Hungary; elek.telek@aok.pte.hu (E.T.); zoltan.ujfalusi@aok.pte.hu (Z.U.); andras.lukacs@aok.pte.hu (A.L.); 2Szentágothai Research Centre, Virological Research Group, University of Pécs, Ifjúság Str. 20, H-7624 Pécs, Hungary; kemenesi.gabor@gmail.com (G.K.); brigitta.zana@gmail.com (B.Z.); jakab.ferenc@pte.hu (F.J.); 3Faculty of Sciences, Institute of Biology, University of Pécs, Ifjúság Str. 6, H-7624 Pécs, Hungary; 4National Laboratory of Virology, University of Pécs, Ifjúság Str. 20, H-7624 Pécs, Hungary; 5Languages for Biomedical Purposes and Communication, Medical School, University of Pécs, Szigeti Str. 12, H-7624 Pécs, Hungary; gabriella.hild@aok.pte.hu; 6Department of Medical Imaging, Clinical Centre, University of Pécs, Ifjúság Str. 13, H-7624 Pécs, Hungary

**Keywords:** SARS-CoV-2, COVID-19, human whole blood, transferrin, calorimetry

## Abstract

SARS-CoV-2 infections are responsible for the COVID-19 pandemic. Transferrin has been found to explain the link between diseases associated with impaired iron transport and COVID-19 infection. The effect of SARS-CoV-2 on human whole blood was studied by differential scanning calorimetry. The analysis of the thermal transition curves showed that the melting temperature of the transferrin-related peak decreased in the presence of SARS-CoV-2. The ratio of the under-curve area of the two main peaks was greatly affected, while the total enthalpy of the heat denaturation remained nearly unchanged in the presence of the virus. These results indicate that SARS-CoV-2, through binding to transferrin, may influence its Fe^3+^ uptake by inducing thermodynamic changes. Therefore, transferrin may remain in an iron-free apo-conformational state, which depends on the SARS-CoV-2 concentration. SARS-CoV-2 can induce disturbance in erythropoiesis due to toxicity generated by free iron overload.

## 1. Introduction

Severe acute respiratory syndrome coronavirus 2 (SARS-CoV-2), a member of the coronavirus family (*Coronaviridae*), is a new human pathogen that is responsible for the coronavirus pandemic that started in 2019 (COVID-19) [1]. In most cases, SARS-CoV-2 Omicron Bal. 4–5 variant infection causes mild upper respiratory tract symptoms or no symptoms at all, both in vaccinated and non-vaccinated individuals [2], while Delta B.1.6.17, alpha, Beta 1.351, B.1.1.7, and E484K variants can cause severe diseases with inferior tract respiratory involvement, leading to life-threatening multi-organ dysfunctions [1,3,4,5,6,7]. Severe COVID-19 disease has been associated with impaired/abnormal iron metabolism, iron deficiency anemia (IDA), hypercoagulopathy, thrombosis, and ischemic stroke [5,6,8,9,10,11]. As a subsequent complication, an increased incidence of iron toxicity-related liver cancer (hepatocellular carcinoma, HCC) has also been observed [12,13,14,15]. The relationship between COVID-19 infection and its associated pathological conditions has yet to be uncovered.

Ferritin and transferrin are essential components of iron metabolism. Ferritin plays a role in iron storage, and its level may classify the severity of COVID-19 symptoms [16]. Transferrin participates in the transportation of iron to the bone marrow, where hemoglobin production takes place, which is a key component of erythropoiesis [17,18]. An increased level of ferritin and transferrin was found in COVID-19 patients and SARS-CoV-2-infected cells as well [19,20,21], suggesting the involvement of these iron metabolism-related proteins in the development of the disease. Transferrin may be involved in COVID-19-related IDA, hypercoagulopathy, and ischemic stroke. However, the link between transferrin and COVID-19-related impairments is unclear [9,10,22,23]. Transferrin is an iron carrier glycoprotein that binds to cellular transferrin receptors and delivers iron by receptor-mediated endocytosis [24,25].

Transferrin has two lobes that are divided into two structurally different subdomains. Each lobe can bind predominantly one iron ion (Fe^3+^) with different affinity [26,27]. The binding of iron induces a conformational change in transferrin. In an iron-free state, transferrin has an open-conformation (apo-transferrin), while in its iron-bound form, the conformation is more closed (holo-transferrin) [28]. Transferrin can be actively involved in the coagulation cascade as a procoagulant by inhibiting antithrombin and factor XIIa [29]. A recent study showed that the expression of transferrin increased with age, and its level was consequently higher in males. As the antithrombin level is not affected by age or gender, the ratio of transferrin/antithrombin is the highest in elder males [23]. These data also support the research finding that the elderly have a higher risk of developing severe COVID-19 disease [30]. All these results suggest that the increased transferrin expression and transferrin/antithrombin ratio may heavily contribute to the development of COVID-19-related coagulopathy with more severe outcomes in older patients. Calorimetric measurements were performed on human whole blood to shed light on how transferrin contributed to the molecular changes caused by SARS-CoV-2. 

## 2. Results

### SARS-CoV-2 Influences the Thermodynamic Properties of Transferrin

To demonstrate the effect of the SARS-CoV-2 virus on blood components, calorimetric measurements were performed on anticoagulated human whole blood samples in the presence and absence of SARS-CoV-2. 

The virus caused significant changes in the thermodynamic properties of blood components. Human blood contains a relatively high number of different proteins, which results in endothermic denaturation peaks that are the collective representations of the individual components (Figure 1 and Figure 3). To overcome this issue, we deconvolved the two main peaks and treated them separately (Figure 2 and Figure 4). The first large endothermic peak (Peak 1) can be mainly attributed to the presence of hemoglobin content in the erythrocytes (T_m_~70 °C), because of its predominancy [31], along with other proteins as well, at low concentrations (Figure 1, Figure 2, Figure 3 and Figure 4), whereas the second large peak (Peak 2) is typically the β-globulin-type transferrin, whose melting temperature (T_m_~80–85 °C) can vary depending on its conformation and iron saturation (~30%) under physiological conditions [26,32,33] (Figure 1, Figure 2, Figure 3 and Figure 4). Binding one metal ion increases the melting temperature significantly, and the incorporation of a second Fe^3+^ shifts the T_m_ of transferrin to even higher temperatures [26]. In addition, other proteins were present at a relatively low concentration, such as γ-globulin (IgG) [32].

The incubation of the samples was performed at different temperatures to see the temperature sensitivity of the virus activity at room temperature (24 °C), body temperature (37 °C), and a simulated high-fever temperature (40 °C). After two hours of incubation, at different temperatures, the shape and the main thermal parameters of the control curves (Figure 5) and the treated samples were the same (Appendix A). Long incubation times (50 h) at all three temperatures showed marked effects of the SARS-CoV-2 virus on the blood components (Figure 3 and Appendix A), while on a shorter time scale (25 h), only the incubation temperature of 37 °C produced changes. Therefore, we focused on the effects of SARS-CoV-2 on human blood samples at 37 °C. At incubation times less than 25 h, no significant difference was found between the control and the virus-containing samples; therefore, the incubation time intervals of 25 h and 50 h were selected to explore the thermal properties of the samples (Figure 1, Figure 2, Figure 3, Figure 4 and Figure 6).

No significant difference was observed between the treated and the control samples in the T_m_ values (melting temperature which belongs to the peak of the endothermic curve where ~50% of the proteins are denatured) of the first peak (Peak 1) at any incubation time (Figure 1, Figure 2, Figure 3 and Figure 4, Figure 6 and Table 1 and Table 2). The other characteristic parameters of Peak 1, for example, the overall shape, the half-width, and the area, remained unchanged. Unlike Peak 1, Peak 2 was very sensitive to both the incubation time and the presence of the SARS-CoV-2 virus. The ratio of the integrated area of Peak 2 to Peak 1 changed dramatically with increasing incubation time (Figure 1, Figure 2, Figure 3 and Figure 4 and Table 1). Mainly, the under-curve area and the T_m_ values of Peak 2 fluctuated, whereas Peak 1 remained constant (Figure 1, Figure 2, Figure 3 and Figure 4). Deconvolutions of the main peaks were performed to make these alterations of the second peak more apparent (Figure 2 and Figure 4). These large differences between the virus-treated and the control samples at longer incubation times indicate that the changes of Peak 2 can be attributed to the SARS-CoV-2 infection.

The most consistent results occurred at 37 °C, which is considered the normal body temperature. No significant differences were found between the virus-treated and the control blood samples when the incubation time was shorter than 15 h (Appendix A). However, drastic changes were seen after 15 h of incubation time (Figure 1 and Figure 3). The melting temperature of Peak 2 increased from 80.69 °C (25 h) to 83.89 °C (50 h) in the non-treated samples, which suggests that transferrin bound more Fe^3+^ ions, which might have been released from other blood sources. The under-curve area also grew considerably in the case of the control samples (Figure 1A, Figure 2A, Figure 3A, Figure 4A and Figure 6), indicating that more energy was required to denature the protein. These results indicate that, after the conformational change, transferrin ended up in a more stable configuration.

The enthalpy change (ΔH) of Peak 2 decreased significantly in the case of the SARS-CoV-2 virus-treated samples compared to the control samples (Table 1). The presence of SARS-CoV-2 seemed to prevent transferrin from being stabilized, which indicates that the virus might have interacted with transferrin and inhibited the uptake of Fe^3+^ by the protein. The same tendency was observed in the enthalpy change, when the incubation temperatures were 24 °C and 40 °C, but the structural changes of transferrin developed after longer incubation times (Appendix A).

VeroE6 (ATCC No. CRL-1586)-type cells were used to replicate SARS-CoV-2 in DMEM (Dulbecco’s Modified Eagle Medium-Merck Life Science Ltd. Október huszonharmadika street 6–10. Budapest H-1117, Hungary). In order to exclude the effects of DMEM (cell debris was removed by centrifugation—see Materials and Methods), separate DSC measurements were carried out. In these tests, the medium was added to the anticoagulated blood samples, and after the given incubation times, the measurements were completed. In the presence of DMEM, the T_m_ and the ΔH values increased (Figure 6, Table 2). These results show that DMEM alone had a very strong and opposite effect on the thermodynamic parameters in the absence of virus, which makes the effect of SARS-CoV-2 on the blood components even more dramatic. Control measurements were also performed with the SARS-CoV-2 virus under the above-described conditions, but with saline solution as a reference. These results provided flat, horizontal DSC curves with no changes (data not shown), which suggests that the presence of SARS-CoV-2 alone does not contribute to the DSC signals.

## 3. Discussion

The 2019 pandemic caused by severe acute respiratory syndrome coronavirus 2 (SARS-CoV-2) can bring about mild upper respiratory tract symptoms, and life-threatening multi-organ diseases as well [1,4,5,6,7]. Although there seems to be a relationship between iron metabolism and COVID-19 infection, the evidence-based pathological background is still unknown. In order to study the effect of SARS-CoV-2 on blood components, calorimetric measurements were carried out with anticoagulated human whole blood samples in the absence and presence of the virus. 

The samples were prepared at an incubation temperature of 37 °C to simulate the effect of SARS-CoV-2 at body temperature. The thermal analysis revealed changes neither in the T_m_ nor in the enthalpy of deconvolved Peak 1 related to the hemoglobin of erythrocytes in the absence and presence of SARS-CoV-2 (Figure 1, Figure 2, Figure 3 and Figure 4, Table 1). Previous studies found that the structure and morphology of red blood cells changed in COVID-19 patients [34,35]. Another scientific report using molecular docking simulation showed a possible interaction between hemoglobin and SARS-CoV-2 [36]. These experimental data suggest that SARS-CoV-2 cannot influence the thermodynamic properties of the predominant hemoglobin (Peak 1), and may not infect the hemoglobin itself; however, the morphology of red blood cells can be modified by the virus. The thermal analysis of deconvolved Peak 2 (transferrin) revealed a substantial decrease of 6.4% in the T_m_ value (ΔT_m_ = 5.16 °C), and a decline of 18.80% in the ΔH value, when the samples were treated with SARS-CoV-2 and incubated for 25 h (Figure 1B and Figure 2B, Table 1). The T_m_ values of Peak 2 of the virus-treated samples were nearly the same as the untreated ones’ values after 50 h of incubation, while the ΔH value decreased by 16.50% in the presence of SARS-CoV-2, which was the same tendency that was observed after a shorter incubation time (Figure 3B and Figure 4B, Table 1).

The change in the T_m_ values of Peak 2 suggests that a more compact and stable conformation of Fe^3+^-bound transferrin formed after a longer incubation time in both the presence and the absence of the virus. This finding can indicate that after over 50 h of incubation, more Fe^3+^ can be released from hemoglobin at 37 °C and bind to transferrin; therefore, the concentration of Fe^3+^-bound holo-transferrin increases. As transferrin has two binding sites for Fe^3+^, while the incubation time was increased, the saturation of both binding sites increased, which resulted in a higher under-curve area (higher Peak 2:1 ratio) and a higher T_m_ value for Peak 2. SARS-CoV-2 could inhibit, at least partially, the uptake of Fe^3+^, when the virus was present for a long time, as mainly the ΔH was sensitive to its presence (Table 1). T_m_ did not change markedly, since the holo-transferrin level was supposedly high, which masked the interaction of SARS-CoV-2 with apo-transferrin. At a shorter incubation time, both the T_m_ and the ΔH values of Peak 2 decreased in the presence of the virus (Table 1). The virus undeniably decreased the thermal stability of transferrin, and the energy consumption of the denaturation was also lower in the presence of the virus. The effect of SARS-CoV-2 could be observed more markedly after a shorter incubation time, when the less rigid iron-free apo-transferrin seemed to be predominant with a significantly lower melting temperature and a lower under-curve area (Peak 2:1 ratio). When the samples were treated for too long, the T_m_ values became nearly insensitive to the presence of SARS-CoV-2 (Table 1). Our findings are in accordance with previously published data, where DSC measurements of iron-titrated transferrin resulted in similar T_m_ values, depending on the occupied Fe^3+^ binding sites [26]. 

Control measurements were performed to exclude the effect of the virus buffer (DMEM) on the human whole blood components. After treating the samples with DMEM exclusively, the analysis showed no significant change in the T_m_ values of Peak 1 and Peak 2, while the under-curve area was considerably higher after 25 and 50 h of incubation compared to the samples where the DMEM was present in a smaller amount. When the virus was not present, the opposite tendency in the change of ΔH suggested an even stronger effect of SARS-CoV-2 on transferrin, and, thus, confirmed our experimental findings (Figure 6, Table 2).

Iron is essential for the human body, and its regulation requires the precisely coordinated functioning of many proteins (e.g., transferrin, ferroportin, ferritin, lactoferrin, hepcidin). One of the most important tasks of this coordinated function is to keep free iron levels in the blood low. Increased serum iron levels generate reactive oxygen species (ROS) and also provide an ideal environment for microbial multiplication [37]. Based on previous findings [38], SARS-CoV-2 can influence the transferrin-mediated iron transport processes, which leads to ineffective erythropoiesis that may cause IDA. HCC and organ impairment can also develop as a late complication of iron-toxicity-induced ROS production that is caused indirectly by SARS-CoV-2 [12,13,15,37]. Moreover, the increased level of transferrin and the change in the transferrin/antithrombin ratio [23,30] can lead to the dysfunction of the coagulation system, especially in males, which results in COVID-19-related hypercoagulopathy, thrombosis, and ischemic stroke (Figure 7). The risk of all-cause mortality was found to increase with elevated serum transferrin saturation levels [39]. The schematic model of the effect of SARS-CoV-2 is represented in Figure 8. It is also known that SARS-CoV-2 infection can directly influence the oxygen uptake by destroying the lung tissues and inducing long-term fibrosis in the lung parenchyma [40,41].

Similar to transferrin, lactoferrin can also chelate two ferric ions with very high affinity (though the presence of a carbonate ion is required for stabilizing the iron binding). We can distinguish an iron-free apo-lactoferrin conformation and a highly stable iron-bound holo-lactoferrin, such as in the case of transferrin [42]. Lactoferrin has anti-inflammatory, antiviral, and immunomodulatory activities, and is expressed and secreted by glandular epithelial cells (as well as by neutrophil cells) when they are involved in immune processes [42]. Due to its beneficial effects, lactoferrin is considered an efficient natural weapon against COVID-19 [37], and its antiviral effect was successfully proven experimentally. This protein can exert its antiviral activity by binding directly to the particles of SARS-CoV-2. Lactoferrin molecules can also obscure the host cell receptors of the virus [43]. Clinical studies proved the effectiveness of lactoferrin in the fight against the SARS-CoV-2 virus in asymptomatic and mild-to-moderate COVID-19 patients [44]. However, one has to be extremely cautious with the mode and timing of the administration of this protein [45]. These findings showed that both transferrin and lactoferrin acted against elevated free iron levels. Interestingly, low iron levels can also trigger certain pathological conditions. Clinical studies reported that decreased levels of hemoglobin, free iron, transferrin, and ferritin might result in increased coagulability and indicate iron deficiency anemia, even if the platelet count and coagulation levels fell in the normal range in pediatric patients [46]. Moreover, this increased coagulability can be triggered or amplified by contraceptives (estrogen), which can result in ischemic stroke and/or venous thromboembolism [11].

To sum up, SARS-CoV-2 can attack different organs at various cellular and molecular levels, which makes it a highly effective and dangerous pathogen. In addition, our results corroborate that transferrin can be the missing link between COVID-19-related severe diseases. 

## 4. Materials and Methods

### 4.1. Sample Preparation

Blood was taken from healthy volunteers in light blue-topped Vacutainer^®^ tubes (BD Vacutainer^®^ Citrate Tubes with 3.2% buffered sodium citrate solution) treated with anticoagulants by healthcare professionals and kept on ice. The iron metabolism-related parameters of healthy whole blood are presented in the Appendix A. SARS-CoV-2 virus isolation was performed in a biosafety level 4 (BSL-4) laboratory of the National Laboratory of Virology at the Szentágothai Research Centre, UPMS, Hungary. During the experimental procedure, we used a Hungarian SARS-CoV-2 isolate (isolated on Vero E6 cells, GISAID accession ID: EPI_ISL_483637). Human whole blood samples were infected by mixing 50 µL of isolated SARS-CoV-2 virus in a concentration of 5.62 × 10^6^ TCID50 with 850 µL of whole blood samples in the BSL-4 laboratory. Afterwards, 800 µL of the mixture was pipetted directly into a conventional Hastelloy batch vessel (V_max_ = ~1 mL), and the samples were incubated at room temperature (24 °C), body temperature (37 °C), or 40 °C for 2, 15, 25, or 50 h. The securely sealed batch vessels (containing the samples) were cleaned by a sterilizing agent (Mikrozid^®^ AF liquid—Schülke & Mayr GmbH – Robert-Koch-Street 2, 22851 Norderstedt, Germany) and kept in it until the DSC analysis could be performed.

### 4.2. Differential Scanning Calorimetry

Differential scanning calorimetry (DSC) is a thermoanalytical technique to study the thermodynamic properties of biological and non-biological samples [47,48,49,50,51]. To investigate the effect of SARS-CoV-2 on the human whole blood, thermal denaturation measurements were performed with SETARAM Micro DSC-III calorimeter. Each measurement was performed in the range of 20–100 °C by using a 0.3 K·min^−1^ heating rate. The sample and the reference (normal saline) were balanced with a precision of ±0.05 mg to avoid corrections with the heat capacity of the vessels. A second thermal scan of the denatured sample was carried out for baseline correction. The deconvolution of DSC data, the analysis of the melting temperature (T_m_), and the calculation of the relative enthalpy change (ΔH) from the area under the deconvolved heat absorption curves were analyzed using the OriginLab Origin^®^ 2021 (version 9.8.5.212) software (OriginLab Corporation, One Roundhouse Plaza, Suite 303, Northampton, MA 01060, USA).

## Figures and Tables

**Figure 1 ijms-23-06189-f001:**
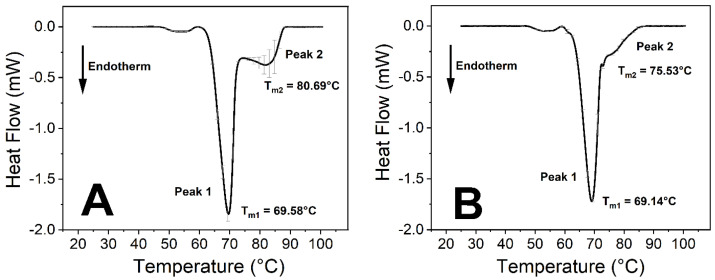
The thermal analysis of untreated and SARS-CoV-2-treated anticoagulated human whole blood samples. (**A**) Control: 25 h of incubation at 37 °C. (**B**) Treated with SARS-CoV-2 and incubated for 25 h at 37 °C. The measurements were performed at least four times independently (*n* ≥ 4) and the results are presented as mean ± SD.

**Figure 2 ijms-23-06189-f002:**
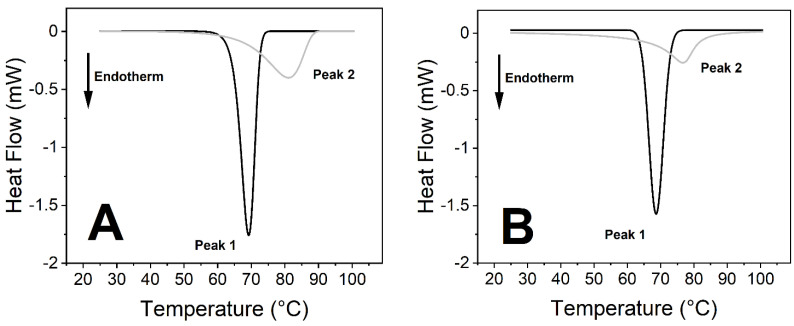
Deconvolution of peak 1 and peak 2 from Figure 1 DSC plots. (**A**) Control: 25 h of incubation at 37 °C. (**B**) SARS-CoV-2 treatment: incubated for 25 h at 37 °C.

**Figure 3 ijms-23-06189-f003:**
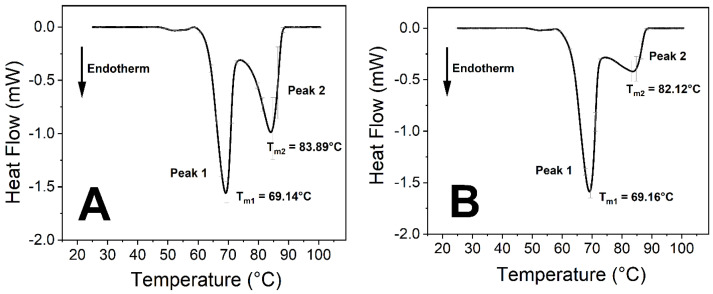
Thermal analysis of untreated and SARS-CoV-2-treated human anticoagulated whole blood samples. (**A**) Control: 50 h of incubation at 37 °C. (**B**) Treated with SARS-CoV-2 and incubated for 50 h at 37 °C. The DSC data represent the mean ± SD of at least four independent measurements (*n* ≥ 4).

**Figure 4 ijms-23-06189-f004:**
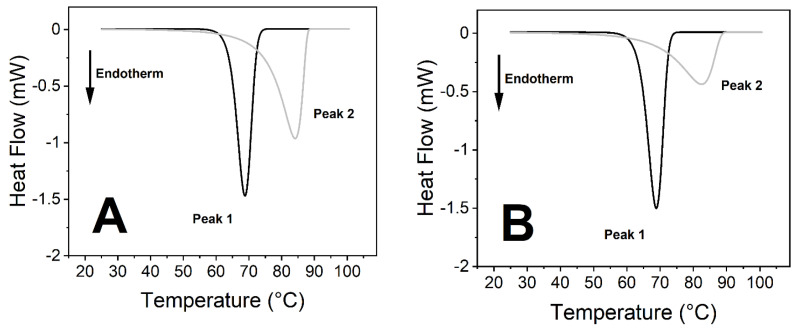
Deconvolution of peak 1 and peak 2 from Figure 3 DSC plots. (**A**) Control: 50 h of incubation at 37 °C. (**B**) Treated with SARS-CoV-2 and incubated for 50 h at 37 °C.

**Figure 5 ijms-23-06189-f005:**
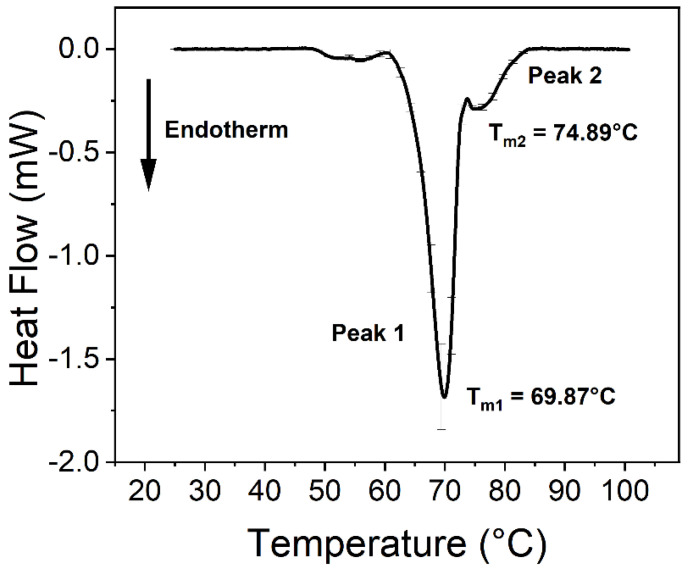
The thermal analysis of untreated anticoagulated whole blood samples after 2 h of incubation. The plot is an average of the same measurements repeated at different incubation temperatures (24 °C, 37 °C, and 40 °C). The DSC data represent the mean ± SD of three independent measurements (*n* = 3).

**Figure 6 ijms-23-06189-f006:**
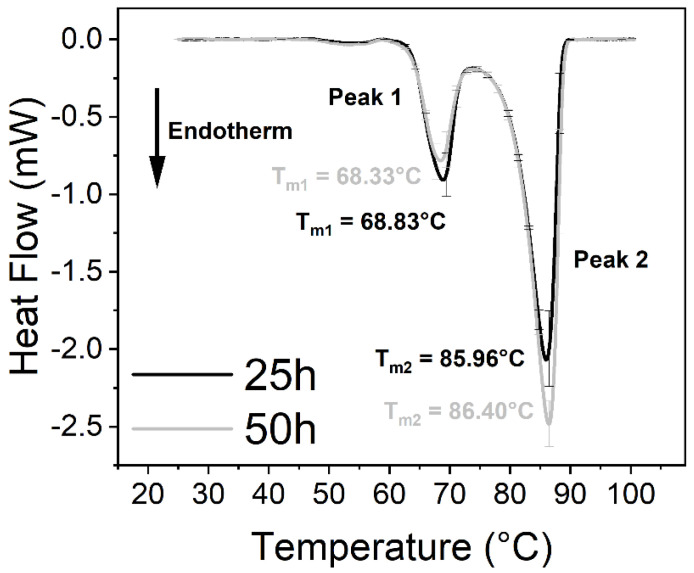
Anticoagulated human whole blood treated with DMEM (Dulbecco’s Modified Eagle Medium) as a control and incubated for 25 h and 50 h at 37 °C. The measurements were performed three times independently (*n* = 3) and the results are presented as mean ± SD.

**Figure 7 ijms-23-06189-f007:**
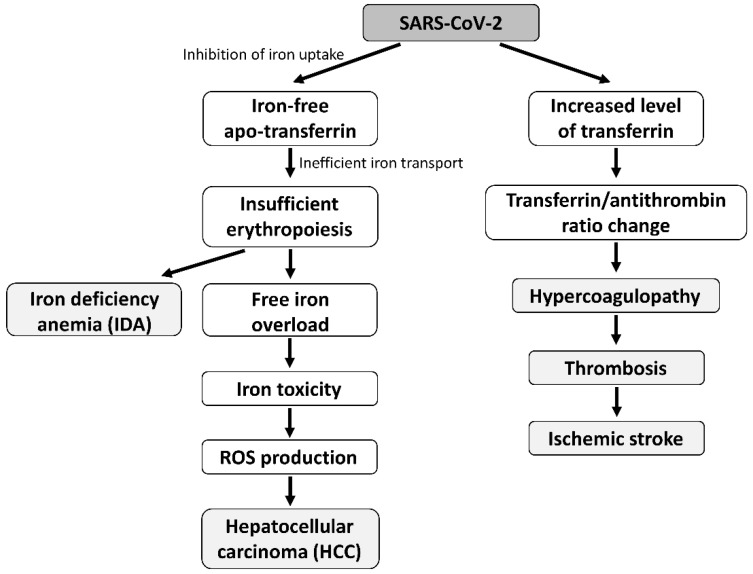
A schematic flow chart model of the possible relationship between the effect of SARS-CoV-2 on transferrin and COVID-19-related severe diseases. The left flow chart represents how SARS-CoV-2 may affect transferrin by inhibiting Fe^3+^ uptake, leading to IDA and HCC. The right flow chart of the model summarizes the literature data of how the increased transferrin level changes the transferrin/antithrombin ratio [29], probably causing hypercoagulopathy, thrombosis, and ischemic stroke.

**Figure 8 ijms-23-06189-f008:**
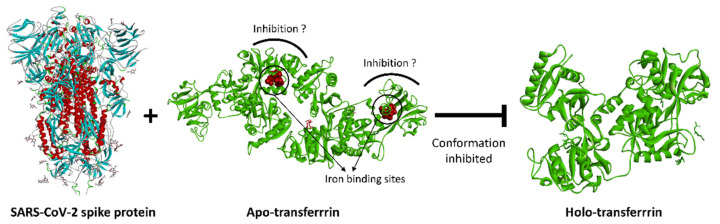
A schematic model of how SARS-CoV-2 (PDB code: 7jwy) may block the uptake of Fe^3+^, resulting in apo-transferrin-like conformation (PDB code: 2hau), maybe leading to the inhibition of holo-transferrin conformation (PDB code: 3v83).

**Table 1 ijms-23-06189-t001:** Thermal characterization of the primary and deconvolved DSC data of human whole blood in the absence and presence of SARS-CoV-2. Data represent mean ± SD (*n* ≥ 4).

Thermal Parameters of Peaks 1 and 2				
	25 h Incubation		50 h Incubation	
	Untreated	SARS-CoV-2	Untreated	SARS-CoV-2
T_m_ of Peak 1 (°C)	69.58 ± 0.11	69.14 ± 0.14	69.14 ± 0.10	69.16 ± 0.09
T_m_ of Peak 2 (°C)	80.69 ± 1.36 *	75.53 ± 0.24 *	83.89 ± 0.58	82.12 ± 1.23
Peak 2:1 ratio ^1^	0.63	0.54	1.25	0.74
Calculated ΔH (J/g)	3.51 ± 0.13 **	2.85 ± 0.05 **	4.12 ± 0.19 **	3.44 ± 0.10 **

^1^ Peak 2:1 ratio was determined from the deconvolved averaged thermal transition. * = statistically significant change between the corresponding control and treated values (*p* < 0.05). ** = very statistically significant change between the corresponding values (*p* < 0.01).

**Table 2 ijms-23-06189-t002:** Thermal characterization of the primary and deconvolved DSC data of control in the presence of DMEM. Mean ± SD (*n* = 3).

Thermal Parameters of Peaks 1 and 2 of DMEM Control		
	25 h Incubation	50 h Incubation
T_m_ of Peak 1 (°C)	68.83 ± 0.17	68.33 ± 0.28
T_m_ of Peak 2 (°C)	85.96 ± 0.24	86.40 ± 0.18
Peak 2:1 ratio ^1^	3.30	4.25
Calculated ΔH (J/g)	4.31 ± 0.06	4.64. ± 0.04

^1^ Peak 2:1 ratio was determined from the deconvolved averaged thermal transition.

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
