# Peer review of "A Possible Way to Relate the Effects of SARS-CoV-2-Induced Changes in Transferrin to Severe COVID-19-Associated Diseases"

_ijms, 2022, doi:10.3390/ijms23116189_

Round 1

Reviewer 1 Report

Dear authors,

I read your manuscript about the effects of SARS-CoV-2 on transferrin and iron metabolism.

  • “In most cases, SARS-CoV-2 infections cause mild upper respiratory tract symptoms or no symptoms at all”, are you talking about omicron Bal.4-5 variant and in vaccinated subject? Delta B.1.6.17, alpha, Beta 1.351, B.1.1.7+E484K variants presented as severe disease and with inferior tract respiratory involvement (Ikea classification). Correct the sentence.
  • Line 74-77, remove from the introduction.
  • Material and method should be move after the introduction. Did you measure the iron, transferrin and ferritin of these patients before the procedure?
  • “ The melting temperature of Peak 2 increased from 80.69 °C (25 hours) to 83.89 °C (50 hours) in the non-treated samples, which suggests that transferrin bound more Fe3+ ions, which might have been released from other blood sources.” Is it statistically significant? P-value?
  • Although a diagram is provided to explain the possible correlation between iron homeostats and the severity of the disease, it is not evident from the discussion section. Furthermore, no clinical implications are reported that can be drawn from this information. To improve discussion read and cite:

Campione, E., Cosio, T., Rosa, L., Lanna, C., Di Girolamo, S., Gaziano, R., Valenti, P., & Bianchi, L. (2020). Lactoferrin as Protective Natural Barrier of Respiratory and Intestinal Mucosa against Coronavirus Infection and Inflammation. International journal of molecular sciences, 21(14), 4903. https://doi.org/10.3390/ijms21144903

Campione, E., Lanna, C., Cosio, T., Rosa, L., Conte, M. P., Iacovelli, F., Romeo, A., Falconi, M., Del Vecchio, C., Franchin, E., Lia, M. S., Minieri, M., Chiaramonte, C., Ciotti, M., Nuccetelli, M., Terrinoni, A., Iannuzzi, I., Coppeda, L., Magrini, A., Bernardini, S., … Bianchi, L. (2021). Lactoferrin Against SARS-CoV-2: In Vitro and In Silico Evidences. Frontiers in pharmacology, 12, 666600. https://doi.org/10.3389/fphar.2021.666600

Campione, E., Lanna, C., Cosio, T., Rosa, L., Conte, M. P., Iacovelli, F., Romeo, A., Falconi, M., Del Vecchio, C., Franchin, E., Lia, M. S., Minieri, M., Chiaramonte, C., Ciotti, M., Nuccetelli, M., Terrinoni, A., Iannuzzi, I., Coppeta, L., Magrini, A., Bernardini, S., … Bianchi, L. (2021). Lactoferrin as Antiviral Treatment in COVID-19 Management: Preliminary Evidence. International journal of environmental research and public health, 18(20), 10985. https://doi.org/10.3390/ijerph182010985

Bonaccorsi di Patti, M. C., Cutone, A., Polticelli, F., Rosa, L., Lepanto, M. S., Valenti, P., & Musci, G. (2018). The ferroportin-ceruloplasmin system and the mammalian iron homeostasis machine: regulatory pathways and the role of lactoferrin. Biometals : an international journal on the role of metal ions in biology, biochemistry, and medicine, 31(3), 399–414. https://doi.org/10.1007/s10534-018-0087-5

Rosa, L., Lepanto, M. S., Cutone, A., Siciliano, R. A., Paesano, R., Costi, R., Musci, G., & Valenti, P. (2020). Influence of oral administration mode on the efficacy of commercial bovine Lactoferrin against iron and inflammatory homeostasis disorders. Biometals : an international journal on the role of metal ions in biology, biochemistry, and medicine, 33(2-3), 159–168. https://doi.org/10.1007/s10534-020-00236-2

Mancilha, E., & Oliveira, J. (2021). SARS-CoV-2 association with hemoglobin and iron metabolism. Revista da Associacao Medica Brasileira (1992), 67(9), 1349–1352. https://doi.org/10.1590/1806-9282.20210555

Author Response

Answers to Reviewer 1

We would like to thank you for the valuable feedback and the overall positive assessment of our manuscript. We addressed your concerns raised during the evaluation of our paper. Please find our point-by-point response to your comments below.

Comments and Suggestions for Authors

Dear authors,

I read your manuscript about the effects of SARS-CoV-2 on transferrin and iron metabolism.

  1. “In most cases, SARS-CoV-2 infections cause mild upper respiratory tract symptoms or no symptoms at all”,are you talking about omicron Bal.4-5 variant and in vaccinated subject? Delta B.1.6.17, alpha, Beta 1.351, B.1.1.7+E484K variants presented as severe disease and with inferior tract respiratory involvement (Ikea classification). Correct the sentence.

Thank you for pointing this out.  We have corrected this section about the Omicron and other variants. In addition, we cited further publications for these SARS-CoV-2 variants in this section (rows 41-46).

  1. Line 74-77, remove from the introduction.

Thank you for the suggestion, this unnecessary section has been removed from the introduction as requested.

  1. Material and method should be move after the introduction. Did you measure the iron, transferrin and ferritin of these patients before the procedure?

Thank you for this suggestion. Materials and Methods has now been moved after the introduction.

The volunteers had a complete blood count test before we used their blood for experimental purposes. These blood test showed that the iron levels of the patients were between 20.09 - 23.21 µmol/L; ferritin was between 128 - 159 ug/L and transferrin was between 2.33 - 3.28 g/L, and transferrin saturation was in the range of 28.16 – 34.32 %. All values were within the reference range.

  1. “The melting temperature of Peak 2 increased from 80.69 °C (25 hours) to 83.89 °C (50 hours) in the non-treated samples, which suggests that transferrin bound more Fe3+ ions, which might have been released from other blood sources.” Is it statistically significant? P-value?

Thank you for pointing this out. We performed a student’s T test; and the P-value was 0.1, which shows weaker evidence. That is why we did not state a significant difference here, though more than 3°C difference suggests a decent change, even if it is statistically not significant. Statistical analysis was carried out for the changes in enthalpy in the control and virus-treated samples, since this can be more relevant. The enthalpy change between 25h control and treated sample was found statistically significant (P-value = 0.0038). The enthalpy change between 50h control and treated sample was also statistically significant (P-value = 0.0081). The temperature change between the Tm values of Peak 2 of 25h control and treated samples were significant as well with a P-value of 0.0168. We represented these data in Table 1 and included them in the text as well. (Rows 220-221)

  1. Although a diagram is provided to explain the possible correlation between iron homeostats and the severity of the disease, it is not evident from the discussion section. Furthermore, no clinical implications are reported that can be drawn from this information. To improve discussion read and cite:

Thank you for pointing this out. We agree with this comment. On the basis of your comments, we have made the necessary changes in the Discussion chapter. (Rows 53; 255-264; 302-307; 306-307; 320-340)

Again, we would like to express our gratitude for reviewing our manuscript. We hope that the changes and corrections could improve the quality of our work, and it can make the manuscript suitable for publication in the prestigious journal of International Journal of Molecular Sciences.

Sincerely yours,

Zoltán Ujfalusi

Reviewer 2 Report

The manuscript of ijms-1715157 studied the changes of human whole blood in the presence and the absence of SARS-CoV-2 by DSC. The results indicated that obvious changes observed for the peak 2 in DSC plots for the human whole blood in the presence and the absence of SARS-CoV-2. The results are interesting. However, the results can not support the conclusions since the whole blood is complicated, which can not only include transferrin but a lot of other components. The authors connected the changes of the DSC plots with the transferrin just because the transferrin might be related to the severe COVID-19 disease, which is unconvincing. There are some other suggestions here:
1. In the references 23, 24, 27, 32 and 35, transferrin was studied in serum samples. Why did this study use whole blood, which contains more complex components including blood cells? And did SARS-CoV-2 infect blood cells?
2. Were the changes in DSC plots caused by the infection of SARS-CoV-2 to blood cells or by the components of SARS-CoV-2? Did the authors think about using another virus (not infect human or not cause severe disease) as a reference to exclude the effects of the components of viruses?
3. It says "Peak 1 is mainly due to the presence of albumin..., while Peak 2..." from line 87 to 91 in the manuscript. It is better to explain why did the Peak 1 was ascribed to albumin and Peak 2 to transferrin. The authors referred to reference 27, while reference 27 studied blood serum, which is not so complex as whole blood.
4. All the analyses and conclusions about transferrin (figure 7 and figure 8) are not directly based on the results, but assumed by the authors.

Author Response

Answers to Reviewer 2

We are grateful for your valuable feedback and comments. Your suggestions are highly appreciated, and you can find our answers below.

Comments and Suggestions for Authors

The manuscript of ijms-1715157 studied the changes of human whole blood in the presence and the absence of SARS-CoV-2 by DSC. The results indicated that obvious changes observed for the peak 2 in DSC plots for the human whole blood in the presence and the absence of SARS-CoV-2. The results are interesting. However, the results cannot support the conclusions since the whole blood is complicated, which can not only include transferrin but a lot of other components. The authors connected the changes of the DSC plots with the transferrin just because the transferrin might be related to the severe COVID-19 disease, which is unconvincing.

There are some other suggestions here:

  1. In the references 23, 24, 27, 32 and 35, transferrin was studied in serum samples. Why did this study use whole blood, which contains more complex components including blood cells? And did SARS-CoV-2 infect blood cells?

Indeed, transferrin was studied in serum samples in the cited publications. In our study, to create an experimental environment that is as close to the „in vivo” conditions of SARS-CoV-2 infection as possible, human whole blood was used to analyze the effects of the virus on the blood components, however, it is a more complex system. Several studies mentioned the possible involvement of transferrin in COVID disease. Katie-May McLaughlin and her co-workers published that transferrin could be a missing link in COVID-19-related coagulopathy (Katie-May McLaughlin et al. 2020, Diagnostics, 10, 539; doi:10.3390/diagnostics10080539). Therefore, we tried to focus on transferrin.

Despite the complexity of whole blood, hemoglobin and transferrin levels dominate in the samples. Furthermore, the two major peaks of DSC data were deconvolved by deconvolution analysis, using OriginLab Origin®2021 software. Deconvolution helps to extract a single peak from overlapping thermal transitions. Thereafter, the deconvolved major peaks were analysed to determine the thermodynamic changes of peak 1 and 2. The DSC measurements showed the two peaks assigned to hemoglobin and transferrin proteins, since their Tm values could be identified based on previous literature results. The thermal transition of hemoglobin of erythrocytes occurs at Tm = ~70 °C (James R. Lepock 2004, Methods 35 (2005) 117–125), whereas, the transition of transferrin can take place at higher temperature values, above 80 °C (for example in healthy individuals Tm= ~85 °C in DSC studies according to Todinova et al. 2011, Anal. Chem. 83, 7992–7998). In addition, Tm depends on the iron binding and saturation of the binding sites of transferrin (Benjamin-Rivera et al. Inorganics 2020, 8, 48; doi:10.3390/inorganics8090048).

 The normal transferrin saturation level can be found in the reference range of 16-45% in healthy individuals under physiological conditions. In our volunteers, the mean transferrin saturation was ~32%, which seems to be in accordance with our control results, since the Tm values of transferrin (saturated with iron) were between 80.69 ±1.36 - 83.89 ±0.58 °C. Whereas, previous studies showed that higher transferrin saturation resulted in higher Tm values (= ~85°C) because of an increased concentration of holo-transferrin conformation (Benjamin-Rivera et al. Inorganics 2020, 8, 48; doi:10.3390/inorganics8090048). Altogether, only transferrin (peak 2) has thermal transition between 80-85 °C depending on the iron saturation and its conformation according to the literature, which is clearly separated from peak 1. These data support/demonstrate that whole blood is a complex system and can be used to measure the thermodynamic changes of the predominant transferrin as an individual peak since the change in enthalpy and the Tm values can be correlated with the protein conformation.  

 Literature results also show that SARS-CoV-2 may infect red blood cells (RBCs). Researchers observed that the morphology and structure of RBCs changed in COVID-19 patients (Francesco Misiti 2021, Sports Med Health Sci. 2021 Sep; 3(3): 181–182 and Thomas et al. J. Proteome Res. 2020, 19, 4455−4469). Cosic et al. found a new evidence of how SARS-CoV-2 could infect RBCs, using a biophysical resonant recognition model (Cosic I et al. 2020. Appl Sci. 11(10):4053). Other studies mention that SARS-CoV-2 can be associated with hemoglobin and influence iron metabolism (Mancilha, E., & Oliveira, J. 2021, Revista da Associacao Medica Brasileira (1992), 67(9), 1349–1352. https://doi.org/10.1590/1806-9282.20210555 and Liu & Li COVID-19: Attacks the 1-Beta Chain of Hemoglobin and Captures the Porphyrin to Inhibit Heme Metabolism). Although, Liu & Li performed only bioinformatic molecular docking analysis, no experimental data were included in their paper. In our study, neither the Tm of peak 1 for hemoglobin (~70 °C) nor the enthalpy changed after 25 or 50 hours of incubation, which suggests that SARS-CoV-2 may not affect the conformation and the structure of hemoglobin.

  1. Were the changes in DSC plots caused by the infection of SARS-CoV-2 to blood cells or by the components of SARS-CoV-2? Did the authors think about using another virus (not infect human or not cause severe disease) as a reference to exclude the effects of the components of viruses?

The analysis of the DSC plots (change in the Tm, enthalpy and the ratio of Peak 2:1) shows that only Peak 2 changed, which can be attributed to transferrin based on the literature (Todinova et al. 2011, Anal. Chem. 83, 7992–7998 and Benjamin-Rivera et al. Inorganics 2020, 8, 48; doi:10.3390/inorganics8090048). Some literature data presents that SARS-CoV-2 may infect red blood cells. Studies have revealed that the morphology and the structure of the membrane of red blood cell can change in COVID-19 patients (Francesco Misiti 2021, Sports Med Health Sci. 2021 Sep; 3(3): 181–182 and Thomas et al. J. Proteome Res. 2020, 19, 4455−4469). Cosic et al. found new evidence of how SARS-CoV-2 could infect red blood cells using a biophysical resonant recognition model (Cosic I et al. 2020. Appl Sci. 11(10):4053). In our study, Peak 1 that was mainly related to the hemoglobin of erythrocytes at Tm= ~70 °C (James R. Lepock 2004) showed no change neither in Tm nor in the enthalpy; however, SARS-CoV-2 seemed to be able to infect red blood cells.

In conclusion, we assume that SARS-CoV-2 may have no influence on the components of red blood cells. In addition, no change in the thermodynamic properties of hemoglobin could be observed, but we hypothesize that the components of the virus are responsible for the change of peak 2 that belongs to transferrin, which are in accordance with several studies cited in our literature review. In addition, we assume in our model (Fig. 7) that the spike protein component of the SARS-CoV-2 may be responsible for the possible association to transferrin, because the virus mostly can interact and infect macromolecules via its spikes. However, no further data are available about how SARS-CoV-2 may interact with transferrin. We are grateful for the thought-provoking idea. We are planning to study the relationship between the virus and transferrin in more details in the near future.

 Thank you for suggesting the use of another virus as a control. We did not think of using another virus as a control, but we performed control measurements on samples containing the isolated virus in DMEM, which resulted in a flat DSC spectrum. This means that the components of the virus have no effect on the observed change of the DSC plots of whole blood (data not shown).

Moreover, we performed other control DSC experiments using DMEM medium (in the lack of the virus) with the whole blood samples in order to exclude the effect of the medium. The results showed no thermodynamic changes, confirming our idea that only SARS-CoV-2 can influence the thermodynamic parameters of transferrin (peak 2). Besides the incubation temperature of 37 °C, the measurements were repeated several times at the incubation temperatures of 24 and 40 °C (Supplementary materials, Fig S1 and S2). In each case, only the transferrin (peak 2) transition curve changed, which clearly demonstrates the effect of the virus. All in all, we think that both the virus control (data not shown) and the DMEM control (Fig. 5, Table 2) are sufficient to confirm the effect of SARS-CoV-2.

  1. "Peak 1 is mainly due to the presence of albumin..., while Peak 2..." from line 87 to 91 in the manuscript. It is better to explain why did the Peak 1 was ascribed to albumin and Peak 2 to transferrin. The authors referred to reference 27, while reference 27 studied blood serum, which is not so complex as whole blood.

Thank you for pointing out these mistakes. The section has been modified on the basis of the reviewer’s comments. The melting temperatures of hemoglobin and transferrin have been included and supported by previous literature. (Rows 121-129)

Peak 1 belongs to the hemoglobin content of the erythrocytes and not to albumin. James R. Lepock measured hemoglobin with DSC and observed similar Tm at 72 °C, using 1 K/min scanning rate. In our measurements (using 0.3 K/min), peak 1 related to hemoglobin was between 69-70 °C.

Although Todinova et al. (former reference 27) studied blood serum in which the Tm of transferrin was at ~85 °C (Todinova et al. 2011, Anal. Chem. 83, 7992–7998), we measured very similar Tm values for peak 2 in whole blood samples. It is also important to emphasize that the Tm value of transferrin can be different depending on the bound iron, which can be different in serum and whole blood samples. However, whole blood can provide a more physiological condition. In our study, the Tm of transferrin varied between 80-85 °C at different incubation times (25 and 50 hours), depending on the bound iron content and the transferrin conformation. Previous results demonstrated that the thermal transition of IgG isotope could occur at ~82 °C, but this was only found in Multiple Myeloma patients, where inflammatory processes highly increase the IgG level. However, in healthy individuals this IgG protein has a relatively low level and do not contribute to peak 2, which is the case in our measurements, as the blood was taken from healthy volunteers.

It is important to highlight that Todinova et al. used 0.8 K/min scanning rate in their serum study, which resulted in ~85 °C for the thermal transition of transferrin as we mentioned above, while in our study, the scanning rate was slower, 0.3 K/min. According to Dergez et al., there can be even a ~4 °C decrease in Tm between 0.8 K/min and 0.3 K/min scanning rates.

To sum up, these data correspond to previous results found for hemoglobin and transferrin transition, which supports the idea that peak 1 is assigned to hemoglobin; whereas, peak 2 can belong to transferrin in our whole blood DSC measurements. We did not find any reference that used whole blood DSC analysis focusing on transferrin; therefore, we cited Todinova et al.’s serum study.

  1. All the analyses and conclusions about transferrin (figure 7 and figure 8) are not directly based on the results, but assumed by the authors.

You have raised an important point here. However, our conclusions contain assumptions and schematic models (Fig. 7 and 8) based on our results. We created the conformational schematic model (Fig. 7) and the flow chart (Fig. 8) to explain the possible biological relevance of how the observed thermodynamic changes (different conformations) of transferrin (peak 2) can play role in and may contribute to iron metabolic dysfunctions and clinical diseases. Based on the different conformational states of transferrin (apo-, and holo-transferrin) found in previous studies, distinct thermodynamic properties can be observed in the absence or presence of iron bound to transferrin. Most importantly, the change in the Tm indicates either the presence of the less rigid apo-transferrin or the more rigid holo-transferrin at even an intermediate conformation, depending on the iron saturation of the two iron-binding sites of transferrin (Benjamin-Rivera et al. Inorganics 2020, 8, 48; doi:10.3390/inorganics8090048).

Altogether, we created the models in Fig. 7 and 8 in order to give a broad overview of the possible relationships between transferrin and its dysfunctions. Furthermore, we explained the relationship between transferrin and these iron based metabolic dysfunctions in the Discussion to present the possible clinical aspects of our thermoanalytical results.

Again, we would like to express our gratitude for reviewing our manuscript. We hope that the changes and corrections could improve the quality of our work, and it can make the manuscript suitable for publication in the prestigious journal of International Journal of Molecular Sciences.

Sincerely yours,

Zoltán Ujfalusi

Round 2

Reviewer 1 Report

Dear Authors,

all the corrections have been made. About "Did you measure the iron, transferrin and ferritin of these patients before the procedure?" thanks for the answer.  You should include iron, transferrin and ferritin mean value in supplementary files. 

Author Response

Answer to Reviewer 1

We would like to thank you for the valuable feedback and suggestion. Please find our response to your comments below.

Comments and Suggestions for Authors

Dear Authors,

all the corrections have been made. About "Did you measure the iron, transferrin and ferritin of these patients before the procedure?" thanks for the answer.  You should include iron, transferrin and ferritin mean value in supplementary files..

Thank you very much for the suggestion. We added a table containing the required parameters in Supplementary Materials (Table S1). These data are now mentioned in the main text as well (rows 82-84).

Extensive English revision has also been performed to improve the quality of the manuscript.

Again, we would like to express our gratitude for reviewing our manuscript. We hope that the changes and corrections could improve the quality of our work, and it can make the manuscript suitable for publication in the prestigious journal of International Journal of Molecular Sciences.

Sincerely yours,

Zoltán Ujfalusi

Reviewer 2 Report

The authors responsed all questions.

Author Response

Thank you very much for the supporting feedback.

Sincerely yours,

Zoltan Ujfalusi